# High Plasma Levels of Neopterin Are Associated with Increased Mortality among Children with Severe Malaria in Benin

**DOI:** 10.3390/diagnostics13030528

**Published:** 2023-01-31

**Authors:** Samuel Odarkwei Blankson, Lauriane Rietmeyer, Patrick Tettey, Liliane Dikroh, Bernard Tornyigah, Rafiou Adamou, Azizath Moussiliou, Caroline Padounou, Annick Amoussou, Benedicta Ayiedu Mensah, Maroufou J. Alao, Gordon Awandare, Nicaise Tuikue Ndam, Christian Roussilhon, Rachida Tahar

**Affiliations:** 1MERIT, IRD, Université de Paris Cité, 75006 Paris, France; 2West African Centre for Cell Biology of Infectious Pathogens, Department of Biochemistry, Cell and Molecular Biology, College of Basic and Applied Sciences, University of Ghana, Accra LG 54, Ghana; 3Department of Immunology, Noguchi Memorial Institute for Medical Research, University of Ghana, Accra LG 54, Ghana; 4Institut de Recherche Clinique du Benin (IRCB), Calavi, Benin; 5Centre Hospitalier Universitaire de l’Oueme, Porto-Novo, Benin; 6Service de Pédiatrie, Centre Hospitalo-Universitaire, Suruléré (CHU-Suruléré), Cotonou, Benin; 7Département de Pédiatrie, Hôpital Mère-Enfant la Lagune (CHUMEL), Cotonou, Benin; 8Department of Parasitology, Noguchi Memorial Institute for Medical Research, University of Ghana, Accra LG 54, Ghana; 9Institut Pasteur, 28 Rue du Docteur Roux, 75015 Paris, France

**Keywords:** neopterin, severe malaria, cerebral malaria, anemia, *Plasmodium falciparum*, Benin

## Abstract

Among the barriers to accessing adequate treatment and high-level monitoring for malaria febrile patients is the lack of effective prognostic markers. Neopterin, which is a marker of monocyte/macrophage activation, was found have increased during severe malaria. In this study, we used quantitative ELISA in order to assess the levels of plasma soluble neopterin in 151 patients from a cohort of Beninese children with severe malaria. We evaluated the prognostic accuracy of this molecule in order to predict the outcome of the disease. Our results show that neopterin levels were not significantly different between patients with different forms of severe malaria, including severe non-cerebral malaria (SNCM) and cerebral malaria (CM). However, the levels of this molecule were found to be higher in patients with severe malarial anemia (SMA) among both CM and SNCM cases (*p*-value = 0.02). Additionally, the levels of this molecule were found to be higher in patients who died from these pathologies compared to those who survived among the two clinical groups (*p*-value < 0.0001) and within the same group (*p*-value < 0.0001 for the CM group, *p*-value = 0.0046 for the SNCM group). The AUC-ROC for fatality among all the severe cases was 0.77 with a 95%CI of (0.69–0.85). These results suggest that plasma neopterin levels constitute a potential biomarker for predicting fatality among severe falciparum malaria patients.

## 1. Introduction

The immune activation marker neopterin is produced by activated monocytes or macrophages when stimulated by IFN-γ, which is released by T-helper cell 1 of the cellular immune response. Neopterin has been found to be essential in inflammatory diseases and apoptosis [1,2]. Neopterin is a by-product of redox reactions and acts as a pro-oxidant that is essential for nitric oxide synthesis, and for the formation of reactive oxygen metabolites [2,3,4,5]. Neopterin is biosynthesized from 2-amino-4-oxo-6 (D-erythro-1′,2′,3′-trihydroxypropyl)-dihydropteridine(D-erythro-neopterin), which is released by activated macrophages after a series of phosphatase action and oxidation steps [2,6]. This pteridine is found in various bio-fluids: serum, plasma, urine, cerebrospinal fluid, ascitic fluid, ovarian cyst fluid, synovial fluid, and saliva; and measurements are performed using Enzyme Linked Immunosorbent assays (ELISAs), Radioimmunoassays (RIAs), and high pressure liquid chromatography (HPLC) [5,7,8,9,10,11,12,13,14,15]. This macrophage activation marker interacts with the pro-inflammatory cytokines of the human host in response to invading pathogens [16]. Its levels were elevated in many severe, neoplastic, and inflammatory diseases, including autoimmune diseases, allografts, parasitic or viral infections, and sepsis [6]. It has been established that increased neopterin concentrations were associated with endothelial damage and the impairment of alveolar epithelial function in patients with lung inflammatory disorders [2,6]. Moreover, serum neopterin levels was identified as a predictor of a patient’s response to treatment and a potent marker of death from acute infections or chronic diseases [1,17,18,19,20,21]. Among the identified inflammatory markers, the diagnostic accuracy of neopterin was evaluated as a potential marker for the severity of malarial disease in Cameroonian children and in travelers with imported malaria. In these studies, neopterin levels on admission were found to be significantly higher in patients with severe *falciparum* malaria compared to those with uncomplicated malaria. Similar results were observed in patients experiencing their first malaria infection compared to patients with a history of malaria. These levels were inversely correlated with the number of malaria attacks [7,22]. This trend was also observed during acute cytomegalovirus infection, where neopterin levels were found to be higher in early infection when compared to both late infections and carrier states [23].

Furthermore, Neopterin level and anemia severity were correlated in Zambian children with either severe or cerebral malaria [24]. Similarly circulating neopterin and Hb levels were inversely correlated in Ghanaian children with varying severities of malaria clinical conditions [25].

However, our study sought to investigate the potential of Neopterin as a reliable biomarker for accurately predicting the prognosis of patients who might diefrom severe malaria. Therefore, we assessed the levels of plasma neopterin in children with various clinical manifestations of severe malaria on admission to hospital. 

## 2. Materials and Methods

### 2.1. Ethics, Study Design, Patient Characteristics, and Sampling

Ethical approval for this study was obtained from the institutional ethics committee of the Research Institute of Applied Biomedical Sciences, Cotonou, Benin (authorization no. 006/CER/ISBA/12) and the Comité National d’Ethique pour la Recherche en Santé (CNERS), Cotonou, Benin. Ethical clearance was obtained from No 50, 25 October 2017, IRB00006860.

Prior to children inclusion to the study, permission was sought from parents or guardians by the signing of a consent form. Participants were free to opt out of the study at any time point if they felt the need to do so without any coercion.

This was a cross-sectional study that was conducted in Cotonou and Porto-Novo from June to September 2012; May to July 2013; and then from December 2017 to July 2019. Cotonou and Porto-Novo, two cities located in a subtropical region with two rainy seasons, have been identified as *P. falciparum* malaria endemic areas with a heterogeneous transmission rate which may exceed thirty infective bites per person each year [26]. The samples were collected from four different hospitals in Cotonou, including Centre Hopitalier Universitaire Mère-enfant de la Lagune (CHUMEL), Centre National Hospitalier Universitaire Hubert Koutoucou Mega (CNHU-HKM), Centre-Hospitalier-Universitaire of Suruléré, and Ménontin Hospital and Oueme/Plateau Hospital in Porto-Novo, Benin.

Children under six years old were recruited after being tested positive to *P. falciparum* first by using a rapid diagnostic test (DiaQuick-Malaria-P. falciparum-Cassette, Dialab; Hondastrasse, Austria), then by performing a confirmatory microscopic examinations for *P. falciparum* mono-infection. The World Health Organization (Geneva, Switzerland) definition criteria for malaria were used to diagnose and categorize patients with varying degrees of severe *P. falciparum* malaria [27]. Two groups of severe malaria patients were defined based on the clinical symptoms at the time of hospital admission. The cerebral malaria group (CM) was defined as having a Blantyre Coma Score (BCS) < 3, characterized by a deep level of unconsciousness with the inability to respond to a painful stimulus, poor verbal response and eye movements; and excluding other causes of coma. The severe non-cerebral malaria (SNCM) group had a BCS > 2, with a better motor and verbal response as well as eye movements, but presented severe malaria symptoms requiring hospitalization and intensive care. This group was characterized by one or more of the following symptoms: pulmonary edema, acute respiratory distress syndrome, hemoglobinuria, acute kidney failure, abnormal liver function, or severe anemia (hemoglobin (Hb) levels under 5 g/dL). Both the CM and SNCM groups included children with or without severe malarial anemia, a common pheneomenon with severe forms of malaria. With permission, 2 to 4 mL of venous blood were collected into citrate phosphate dextrose adenine (CPDA) tubes. The plasma from the blood samples was then stored at −80 °C for the ELISA test after 10 min of centrifugation of the whole blood at 1500× *g* at 4 °C. Once collected, patients’ personal and clinical data were included in an ad-hoc data form on MS-Excel. The data was anonymized by replacing individual names with unique codes for privacy, thus, preventing bias in data analysis.

### 2.2. Competitive ELISA Method for the Determination of Neopterin in Plasma

Neopterin levels were assessed using a competition ELISA test from IBL International (Hamburg, Germany) using 20 µL of undiluted samples. The competition between a peroxidase-conjugated and non-conjugated antigen was evaluated for a fixed number of coated anti-neopterin antibody binding sites. The unbound antigen was removed by washing, and the optical density (OD) was measured after substrate reaction. When the obtained OD values were outside the standard reference range, the dilutions were modified accordingly.

### 2.3. Statistical Analysis

Statistical analyses were conducted using GraphPad Prism v9. The median and 95% Confidence Intervals (CIs) of the neopterin levels were compared between the two clinical groups (CM versus SNCM) for the patients who survived versus those who died, and for the patients with severe malarial anemia (SMA) versus the patients without cases of anemia (NSMA) using the Mann–Whitney U test. The validity of neopterin as a biomarker for malaria-related severity, death, and severe anemia was assessed using a receiver operating characteristic (ROC) curve analysis. This is a two-dimensional measure of classification performance where the area under the ROC curve (AUC) accurately measures discrimination, i.e., reflects the power of a quantified parameter in order to distinguish between two clinical groups. The greater the AUC, the better the test.

## 3. Results

### 3.1. Patient Characteristics

A total of 151 childrenwere diagnosed with severe malaria. Seventy-six (50.3%) of them had CM while 75 (49.7%) had SNCM. There were no significant differences in age, sex, temperature, parasitemia, or hemoglobin levels, neither for most of the other clinical parameters determined for the patients from either groups (Table 1). Obviously, as expected, there was some significant difference (<0.0001) in the presence of impaired consciousness in the CM group, as measured by a BCS < 3, which was the main criterion used during the initial clinical examination to differentiate between CM and SNCM patients. The fatality rate, reflected by the death to survival ratio, was 27 to 49 (35%) for the CM group and 6 to 69 (8%) for the SNCM group (Table 1). Therefore, the percentage of patients with fatal evolution was 4.4-fold higher in the CM group than in the SNCM group (*p*-value < 0.0001).

### 3.2. Neopterin Concentrations 

In total, 151 plasma samples were available for the neopterin measurements. Seventy-six of such samples were obtained from the CM group and 75 samples were obtained from SNCM patients. Our results showed no significant difference between the medians of the neopterin concentrations for the two groups; the median and 95% CI for the CM group was 43.25 and (41.23–71.63) ng/mL, respectively, and the median and 95% CI for the SNCM group was 42.23 and (39.51–54.1) ng/mL, respectively (Figure 1A).

However, the neopterin concentrations were higher in the children who died, with a median of 65.83 and a 95%CI of (56.84–119.9) ng/mL, than concentrations in those who survived, with a median of 37.67 and a 95% CI of (36.06–46.7) ng/mL, (*p*-value < 0.0001). This result was also confirmed within each clinical group (58.64 (55.63–90.78) ng/mL and 30 (30.18–44.73) ng/mL for the CM and SNCM groups, respectively) (Figure 1(B.1,B.2)).

Furthermore, the ROC analysis shows a value of the Area Under the Curve of the Receiver Operating Characteristic (AUC-ROC) of 0.77 (0.69–0.85) ng/mL for fatality (Figure 1C).

Furthermore, we also found that the neopterin concentrations were higher in children who had severe malarial anemia (SMA), with a level of Hb < 5 g/dL, (50.1 (46.29–63.86) ng/mL) compared to those with a level of Hb > 5 g/dL (35.82 (36.16–50.32) ng/mL), (*p*-value = 0.022) (Figure 2A).

Among the 76 children who had CM, 35 had SMA and 41 had Hb levels > 5g/dL. Similarly, among the 75 patients from the SNCM group, 32 of them had SMA and 38 had Hb levels > 5g/dL. The AUC-ROC value shows that neopterin is weak at distinguishing between children with SMA and those without non-severe malarial anemia (NSMA) (AUC = 0.6030, (0.51–0.69) ng/mL) (Figure 2B).

## 4. Discussion

The challenges in the clinical management of severe malaria partially result from the lack of reliable prognostic markers for differentiating patients who are at risk of death at admission, and for allowing for prompt adequate treatment and special attention to be given from health care providers [28].

It has been shown that neopterin concentrations increase during acute infections, including severe malaria, highlighting the diagnostic value of this molecule as a marker of severity [7,22,27,28,29]. Capitalizing on this finding, our study further investigated the potential of neopterin to differentiate between patients with cerebral malaria and patients with the clinical symptoms of severe malaria at admission. In addition, our study aimed to predict which group was at the greater risk of poor outcome. 

Analyzing our results, we observed from Table 1 that both groups had high body temperatures (>37 °C) indicative of fever, hypoglycemia (a blood sugar level of <2.2 mmol/L), which is a risk factor for mortality, as well as high parasitemia (>100,000 parasites/µL), which is indicative of severe malaria [30]. There was no significant difference in Hb levels between the CM and SNCM groups (*p*-value = 0.6712). Although a rise in mortality was observed at Hb levels below 3 g/dL during admission [31,32], we did not observe any significant difference in mortality between children with SMA and those with Hb levels > 5.

The neopterin concentrations of those in the CM group did not differ from those in the SNCM group, as shown in (Figure 1A), as both CM and SNCM are classified as severe forms of malaria, characterized by high immune activation and inflammation in response to the harmful effects of parasitized red blood cells circulating in the body and in the endothelial microvascular beds [33]. According to Picot, some increased levels of serum neopterin indicate the intensity of cellular immunity activation [8]. Furthermore, Brown et al. show that increased levels of IFN-γ in patients with acute *P. falciparum* malaria are positively correlated with neopterin levels [34].

Interestingly, our results showed significantly higher levels of neopterin concentrations in patients who died compared to patients who survived, *p*-value < 0.0001 among children with CM or SNCM, as well as between the children who died and the children who survived within the same group (*p*-value < 0.0001 and *p*-value = 0.0046 for the CM and SNCM groups, respectively) (Figure 1(B.1,B.2)). These results are also supported by the AUC-ROC curve of 0.77 (Figure 1C), indicating a good predictive value for fatality among children with severe malaria on the day of admission.

The concomitant release of neopterin, along with the inflammatory mediators of the host defense mechanism upon activation of monocytes/macrophages during malaria, is likely to initiate the production of reactive oxygen species (ROS) and reactive nitrogen species (RNS) [16]. Oxidative stress can inhibit intracellular Ca^2+^ transients in various cell types, thereby affecting cellular redox systems [35,36]. One of the actions of neopterin is to stimulate the genetic expression of inducible nitric oxide synthase (iNOS) by inducing the translocation of the transcription factor NF-κB into the nucleus, which leads to the subsequent release of nitric oxide (NO) [37]. NO has vasodilator effects [38]. Additionally, raised NO levels due to increased neopterin levels during a pathologic shock state are likely to increase blood flow while reducing blood pressure in the blood vessels. This increases cytoadhesion and the sequestration of *P. falciparum*-infected erythrocytes, causing hyperinflammation, as well as hypotensive reactions [39,40,41]. Neopterin also stimulates the production of TNF-α [42], a pro-inflammatory cytokine. Its addition to the other inflammatory mediators and existing ROS/RNS could lead to deleterious events over the course of an acute malaria infection. Furthermore, as Hoffman argues, neopterin may regulate the intercellular adhesion molecule-1 (ICAM-1) within the alveolar epithelium. An increase in the expression and release of ICAM-1 in alveolar epithelial cells was observed after incubation with neopterin [6]. ICAM-1 is known to play a major role in severe malaria. Postmortem studies have found that children who died from cerebral malaria had an upregulation of ICAM-1 on brain endothelial cells, which co-localized with infected erythrocytes [43]. These observations may account for the high neopterin concentration in children with severe malaria and, subsequently, for the high level of this molecule found in children who died from this pathology.

Our results also indicate increased neopterin concentrations in children with SMA compared to those with Hb levels > 5 g/dL (*p*-value = 0.02). It is important to note that *P. falciparum* infections accelerate the development of anemia as the hemolysis of both parasitized and non-parasitized red blood cells leads to a decline in hematocrit [44,45,46,47]. Moreover, patients with acute *P. falciparum* malaria develop ineffective erythropoiesis, which results from inflammation induced by IFN-γ implicated in neopterin production [48]. The pro-inflammatory cytokine responsible for the production of neopterin, together with its cascade of inflammatory mediators upon the activation of macrophages, have been implicated in dyserythropoiesis and even in red cell apoptosis in malaria patients [49,50,51,52,53]. This may explain the higher levels of neopterin in children with SMA in our study. Consequently, the nonspecific nature of the production of neopterin decreases its relevance as a biomarker for malaria severity, but could be considered in association with a specific marker of malaria infection. Furthermore, even though we used 151 patients, studies in other patient cohorts with different genetic and immunologic backgrounds are needed in order to confirm our findings.

## 5. Conclusions

We have shown, using 251 patients with severe malaria, that neopterin concentrations were high in both the CM and SNCM groups on the first day of admission to the hospital. These concentrations were higher in children with SMA compared to children with Hb levels > 5, and in children who died compared to those who survived. Neopterin may therefore qualify as a good prognostic biomarker of malaria for fatality in patients. Neopterin assessment at admission may also provide clinicians with firsthand information for rapid management of severe malaria patients if it can be developed as a rapid diagnostic test.

## Figures and Tables

**Figure 1 diagnostics-13-00528-f001:**
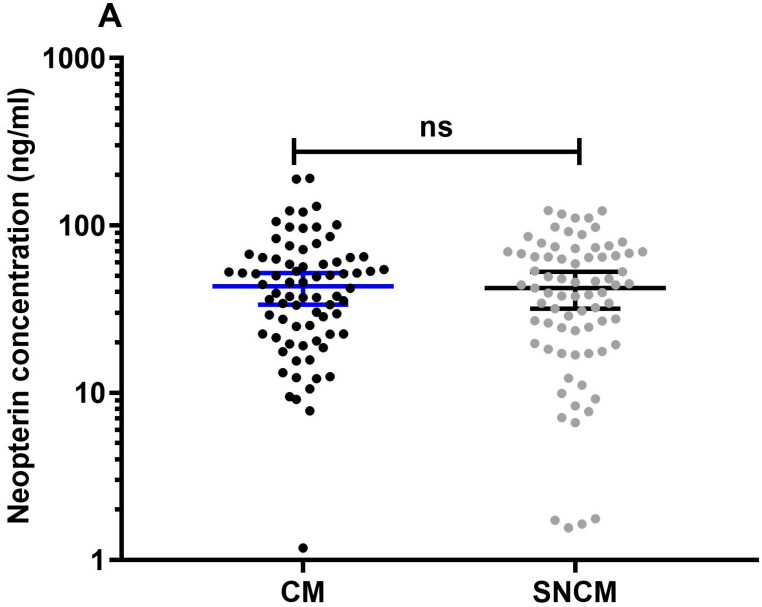
Neopterin concentrations in plasma at admission to hospital (**A**) in children with CM (black dots) compared to children with SNCM (grey dots), (**B.1**) children who died (red dots) compared to those who survived (green squares), (**B.2**) children who died compared to those who survived in the CM group (blue squares) and in the SNCM group (green dots), and (**C**) ROC curve analysis predicting neopterin’s performance in differentiating fatal from non-fatal cases on the day of admission to hospital.

**Figure 2 diagnostics-13-00528-f002:**
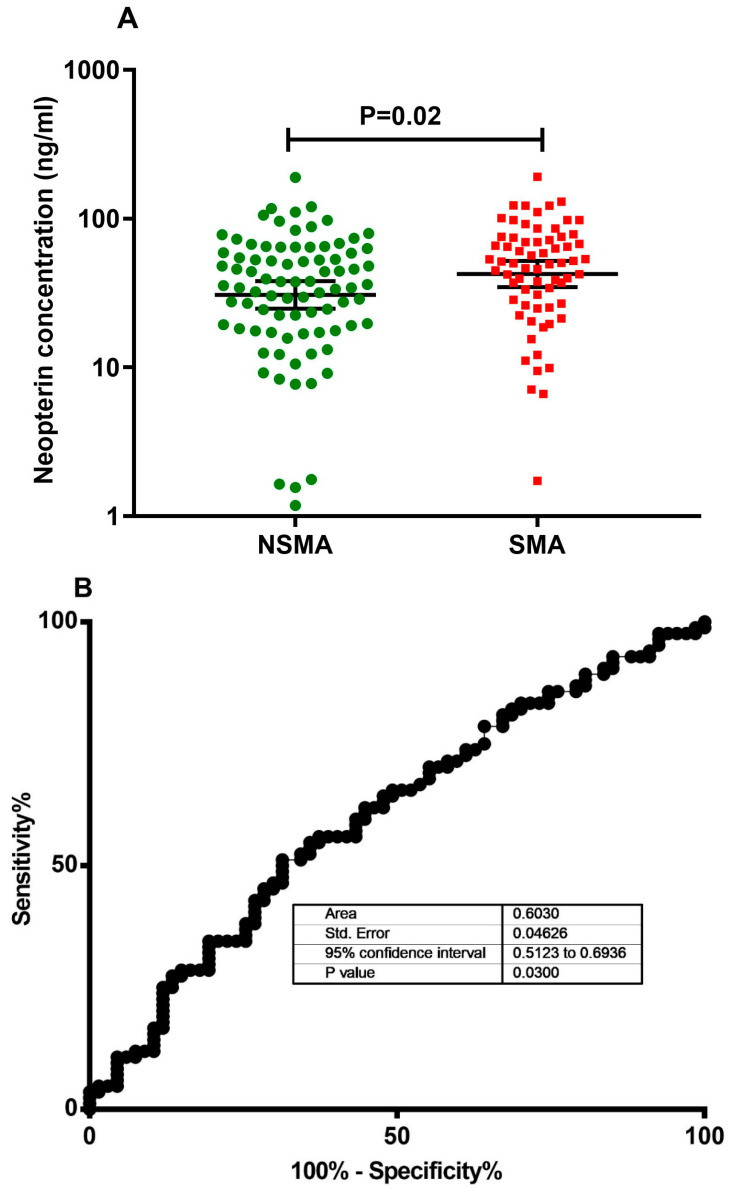
(**A**) Plasma levels of neopterin among children with SMA, as defined by Hb levels < 5g/dL, and children without SMA, as defined by Hb levels > 5g/dL, and (**B**) ROC curve analysis predicting neopterin’s performance in differentiating SMA from NSMA.

**Table 1 diagnostics-13-00528-t001:** Biological, clinical, and parasitological futures of the Beninese children enrolled in the study at admission to hospital. The non-parametric Mann–Whitney U test was used to compare the clinical groups.

	Severe Cerebral Malaria (CM) *n* = 76	Severe non-Cerebral Malaria (SNCM) *n* = 75	*p*-Value
Age (months), (IQR)	39 (26–48) (7–72)	35 (24–48) (6–67)	0.2129
Sex ratio (female/male)	31/45	31/44	0.94
Temperature °C	38.2 (37.5–39) (35.5–40.9) *N* = 72	38.2 (37.6–39) (36–40.6) *N* = 71	0.88
Parasitemia (P/μL) Geo Mean (IQR), (min–max)	139,801 (10,058–173,334) (298–864,000) *N* = 76	278,560 (10,288–298,286) (141–344,000) *N* = 73	0.2511
Hemoglobin (g/dL)	5.8 (4–7.2) (0.8–13.9) *N* = 75	5.6 (3.9–7) (2.1–11.5) *N* = 75	0.6712
MCH pg/L *n* = 61	24.3 (23.4–26.2) [4.9–28] *N* = 44	24.6 (22.2–26.5) (8.4–31.6) *N* = 17	0.8452
VGM *n* = 63	77 (72.4–80.6) (61–91.4) *N* = 47	77.7 (72.2–79.2) (59.4–79.9) *N* = 16	0.6084
leucocytes in blood U/L *n* = 77	13,450 (6775–20,275) (5–50,000) *N* = 58	13,500 (9900–32,300) (12.5–86,100) *N* = 19	0.3834
leucocytes in CSF U/L *n* = 57	5 (2–17.5) (0–249) *N* = 44	6 (1–12.5) (0–91) *N* = 13	0.5554
Glycemia mmol/L *n* = 84	1 (0.7–1.6) (0.1–3.8) *N* = 64	1 (0.9–1.4) (0–16) *N* = 20	0.6040
Proteins (g/L) *n* = 58	0.2 (0.1–0.4) (0.1–1.5) *N* = 42	0.2 (0.1–0.3) (0.03–0.7) *N* = 16	0.2005
Glycose (g/L) in CSF *n* = 64	0.7 (0.5–0.8) (0.3–14.5) *N* = 46	0.9 (0.5–1.0) (0.4–9.5) *N* = 18	0.2270
ABO Rhesus (%)	A+(26.5%), A−(2.9%), B+(32.4%), AB+(2.9%), O+(32.4%), O−(2.9%) (*N* = 34)	A+(46.7%), B+(20%), O+(33.3%) (*N*=15)	
BCS *n* = 151	2 (1–2) (0–2) *N* = 76	5 (3–5) (3–5) *N* = 75	<0.0001
SMA (%) *n* = 151	35/76 (46.1%) *N* = 76	32/75 (42.6%) *N* = 75	0.6754
Deaths (%) *n* = 151	27/76 (35.5%) *N* = 76	6/75 (8%) *N* = 75	<0.0001

Variables are presented as medians with interquartile ranges (IQR), (25th and 75th percentiles) and minimum and maximum values. For parasitemia, the measurements include the geometric mean and the median IQR, minimum, and maximum values. The number of cells were measured for the leucocyte variables. VGM, mean globular volume; MCH, mean cell hemoglobin; CSF, cerebrospinal fluid; BCS, Blantyre coma score.

## Data Availability

The datasets analyzed during the current study are not publicly available but can be provided by the corresponding author on reasonable request.

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
