# Peer review of "High Plasma Levels of Neopterin Are Associated with Increased Mortality among Children with Severe Malaria in Benin"

_diagnostics, 2023, doi:10.3390/diagnostics13030528_

Round 1

Reviewer 1 Report

Journal: Diagnostics

 Manuscript ID: diagnostics-2103840

Title: High plasma Levels of Neopterin are associated with increased mortality among children with severe malaria in Benin

 Authors: Samuel Odarkwei Blankson, Lauriane Rietmeyer, Patrick Tettey, Liliane Dikroh, Bernard Tornyigah, Rafiou Adamou, Azizath Moussiliou, Caroline Padounou, Annick Amoussou, Benedicta Ayiedu Mensah, Maroufou Jules Alao, Gordon Awandare, Nicaise Tuikue Ndam, Christian Roussilhon, Rachida Tahar

 Submitted to section: Diagnostic Microbiology and Infectious Disease

General Comments

In endemic areas, efficient diagnostic tools for the management of severe malaria in hospitalized patients are certainly desirable. Clinical biomarkers also for malaria for their potential in prognosis and predictions may be particularly useful in clinical practice to identify patients at risk for complications, particularly in children.  In view of this, the plasma evaluation of the soluble Neopterin molecule in children with different forms of severe malaria carried out in this cross-sectional study is appreciable and underlines the importance of research on diagnostic/predictive biomarkers in a neglected disease such as malaria. However, the authors should point out that the study has some limitations. The number of severe malaria cases evaluated in the study is not very high, and even if the Neopterin molecule has shown good clinical performance, it is not specific for Plasmodium spp., and like most of the biomarkers it is related to the host response. Therefore, the results of this study require caution in their interpretation and the Authors should mention these limitations in the discussion or conclusion sections.

I suggest this review article to the Authors: Loick Pradel Kojom Foko , Geetika Narang , Suman Tamang , Joseph Hawadak a, Jahnvi Jakhan, Amit Sharma and Vineeta Singh. The spectrum of clinical biomarkers in severe malaria and new avenues for exploration. Virulence, 2022, VOL. 13, NO. 1, 634–653; https://doi.org/10.1080/21505594.2022.2056966

In conclusion, in my opinion the article deserves to be published in Diagnostics after minor revision.

Other specific comments are reported below.

Abstract

Line 29: I suppose the verb is missing in the sentence: “ The AUC-ROC for fatality among all severe cases = 0.77, 95%CI of [0.69 – 0.85].”

 Materials and Methods

Lines 92-94: Please, provide the WHO reference

Line 111: Please, check the company name: “IBL International R”

Line 119: It would be better to write in full: “…Confidence Interval (CI)….

Results

Line 139: Table 1: it would be good rearrange the table to make it easier to read and, please, add the footnotes to explain IQR, TCMH, VGM acronyms. Moreover, what “Geo Mean” means, after Parasitemia (P/µL)?

Line 162-163: This sentence should be moved to the discussion section: “Thus, plasma neopterin levels proved to be a good marker for predicting fatality  among the two severe malaria groups”

Lines 147, 152, 169, 175: Bold on not? “(Figure 1A); (Figure 1 B1, B2);  (Figure 2A); (Figure 2B)

Line 174: The acronym NSMA should be moved in Materials and Methods, line 121.

Discussion

Reference n. 26 is not mentioned.

Conclusion

Too approximate conclusions, please deepen the concepts better (as explained above)

List of abbreviations

Some acronyms are missing

References

Check for typos: line 314; line 366 (Benin, West Africa)

Author Response

Dear reviewer;

We would like to thank you for considering our manuscript and also thank the referees for their significant assessment of our work. We have considered all the remarks and comments raised by the reviewers in order to improve our manuscript. Please find our responses as well as the revised manuscript.

General Comments

  • In endemic areas, efficient diagnostic tools for the management of severe malaria in hospitalized patients are certainly desirable. Clinical biomarkers also for malaria for their potential in prognosis and predictions may be particularly useful in clinical practice to identify patients at risk for complications, particularly in children. In view of this, the plasma evaluation of the soluble Neopterin molecule in children with different forms of severe malaria carried out in this cross-sectional study is appreciable and underlines the importance of research on diagnostic/predictive biomarkers in a neglected disease such as malaria. However, the authors should point out that the study has some limitations. The number of severe malaria cases evaluated in the study is not very high, and even if the Neopterin molecule has shown good clinical performance, it is not specific for Plasmodium, and like most of the biomarkers it is related to the host response. Therefore, the results of this study require caution in their interpretation and the Authors should mention these limitations in the discussion or conclusion sections.

Author responses are in blue

  • We thank the reviewer for the constructing comments made on our manuscript, and we agree to revise our work accordingly.As pointed by the reviewer the low number of individuals used and the nonspecific nature of the neoptrin weaken the conclusion we drawn from our results, thus we added two sentences in the discussion section to introduce the limitations of our study “Consequently, the nonspecific nature of production of the neopterin decrease its alone relevance as biomarker for malaria severity but could be considered in association with a specific marker of malaria infection. Furthermore, even though we used 151 patients, studies in other patient cohorts with different genetic and immunologic backgrounds are needed to confirm our findings.”

  • I suggest this review article to the Authors: Loick Pradel Kojom Foko , Geetika Narang , Suman Tamang , Joseph Hawadak a, Jahnvi Jakhan, Amit Sharma and Vineeta Singh. The spectrum of clinical biomarkers in severe malaria and new avenues for exploration. Virulence, 2022, VOL. 13, NO. 1, 634–653; https://doi.org/10.1080/21505594.2022.2056966

2) The review was cited in discussion section of the manuscript

In conclusion, in my opinion the article deserves to be published in Diagnostics after minor revision.

Other specific comments are reported below

Abstract

Line 29: I suppose the verb is missing in the sentence: “ The AUC-ROC for fatality among all severe cases = 0.77, 95%CI of [0.69 – 0.85].”

Yes, the verb was missing in this sentence we corrected to: The AUC-ROC for fatality among

all severe cases was 0.77, 95%CI of [0.69 – 0.85]

 Materials and Methods

Lines 92-94: Please, provide the WHO reference:

We provided the corresponding reference

Line 111: Please, check the company name: “IBL International R”

We checked for the company name and we corrected to IBL International

Line 119: It would be better to write in full: “…Confidence Interval (CI)…

We corrected CI to Confidence Interval (CI)

Results

Line 139: Table 1: it would be good rearrange the table to make it easier to read and, please, add the footnotes to explain IQR, TCMH, VGM acronyms. Moreover, what “Geo Mean” means, after Parasitemia (P/µL)?

We added a footnotes to explain the abbreviations

[Geo Mean]: Geometric Mean,  [IQR]: 75% and 25% Interquartile,  VGM:Mean Globular Volume,  MCH: Mean Cell Haemoglobin. We also added these terms in the abbreviation section.

Line 162-163: This sentence should be moved to the discussion section: “Thus, plasma neopterin levels proved to be a good marker for predicting fatality among the two severe malaria groups”

The sentence is already present in the discussion section, therefore we rephrased this sentence in the results section to “Furthermore, ROC analysis show a value of the Area Under the Curve of Receiver Operating Characteristic (AUC-ROC) of 0.77 [0.69 – 0.85] ng/ml for fatality (Figure 1C)”.

Lines 147, 152, 169, 175: Bold on not? “(Figure 1A); (Figure 1 B1, B2); (Figure 2A); (Figure 2B)”

We made Figures in bold to according to the journal instructions

Line 174: The acronym NSMA should be moved in Materials and Methods, line 121.

We moved the abbreviation NSMA to Materials and Methods

Discussion

Reference n. 26 is not mentioned.

We changed reference Nahum et al to a more recent adequate reference Tokponnon et al 2023: Entomological Characteristics of Malaria Transmission across Benin: An Essential Element for Improved Deployment of Vector Control Interventions; Insects 10.3390/insects14010052

Conclusion

Too approximate conclusions, please deepen the concepts better (as explained above)

We have chosen to limit our conclusion to the results of the study with the objectives to not extrapolate and being concise.

We change the conclusion to: We have shown that neopterin concentrations are high in both CM and SNCM groups on the first day of admission to the hospital. These concentrations were higher in children with SMA than children with Hb >5 as and in children who died compared to those who survived. Therefore, neopterin presents a good prognostic biomarker of malaria for fatality in patients and its assessment at admission to hospital in malaria confirmed patients could provide clinicians with a firsthand information for rapid management of severe malaria patients if it can be developed into a rapid diagnostic test.

List of abbreviations

Some acronyms are missing

We added the missing Acronyms:

References

Check for typos: line 314; line 366 (Benin, West Africa)

We corrected to Benin, West Africa, and the typos error in line 314

Reviewer 2 Report

This is an interesting and carefully performed study demonstrating a correlation between plasma levels of neopterin and mortality in children with severe malaria. It has been known for some time that neopterin is elevated in severe malaria. The association between neopterin and mortality is original to this manuscript. The limited capacity of neopterin to distinguish between patients with severe and nonsevere malarial anemia, and the lack of a difference between cerebral and non-cerebral malaria are also moderately novel findings.

The methodologies used and the analysis performed appear to be appropriate and are well described.

There are some minor issues of capitalization consistency, particularly in the title and possibly in the institutional affiliations of the authors that require review.

The term "plasma" should be used instead of "plasmatic"

In the description of patient consent, it should be clarified whether or not permission was "sought" or whether permission was "obtained" from parents.

Author Response

Dear reviewer;

We would like to thank you for considering our manuscript and also thank the referees for their significant assessment of our work. We have considered all the remarks and comments raised by the reviewers in order to improve our manuscript. Please find our responses as well as the revised manuscript.

Comments and Suggestions for Authors

1) This is an interesting and carefully performed study demonstrating a correlation between plasma levels of neopterin and mortality in children with severe malaria. It has been known for some time that neopterin is elevated in severe malaria. The association between neopterin and mortality is original to this manuscript. The limited capacity of neopterin to distinguish between patients with severe and nonsevere malarial anemia, and the lack of a difference between cerebral and non-cerebral malaria are also moderately novel findings.

The methodologies used and the analysis performed appear to be appropriate and are well described.

There are some minor issues of capitalization consistency, particularly in the title and possibly in the institutional affiliations of the authors that require review.

Author response

1) We reviewed the document and corrected the capital consistency

The term "plasma" should be used instead of "plasmatic"

Author response

2) We changed plasmatic to Plasma

In the description of patient consent, it should be clarified whether or not permission was"sought" or whether permission was "obtained" from parents.

Author response

3) The patients consent was sought and obtained, we corrected the sentence to: Before children were included to participate in the study, parents or guardians have agreed and signed a consent form.
